# *retro*-Tango enables versatile retrograde circuit tracing in *Drosophila*

**Altar Sorkaç[1,2], Rareş A Moşneanu[1,2], Anthony M Crown[1,2], Doruk Savaş[1,2], Angel M Okoro[1,2], Ezgi Memiş[1,2], Mustafa Talay[1,2†], Gilad Barnea[1,2]\***

[1]Department of Neuroscience, Brown University, Providence, United States; [2]Carney Institute for Brain Science, Brown University, Providence, United States

**Abstract:** Transsynaptic tracing methods are crucial tools in studying neural circuits. Although a couple of anterograde tracing methods and a targeted retrograde tool have been developed in *Drosophila melanogaster*, there is still need for an unbiased, user-friendly, and flexible retrograde tracing system. Here, we describe *retro*-Tango, a method for transsynaptic, retrograde circuit tracing and manipulation in *Drosophila*. In this genetically encoded system, a ligand-receptor interaction at the synapse triggers an intracellular signaling cascade that results in reporter gene expression in presynaptic neurons. Importantly, panneuronal expression of the elements of the cascade renders this method versatile, enabling its use not only to test hypotheses but also to generate them. We validate *retro*-Tango in various circuits and benchmark it by comparing our findings with the electron microscopy reconstruction of the *Drosophila* hemibrain. Our experiments establish *retro*-Tango as a key method for circuit tracing in neuroscience research.

**\*For correspondence:**
gilad_barnea@brown.edu

**Present address:** [†]Howard Hughes Medical Institute, Department of Molecular and Cellular Biology, Harvard University, Cambridge, United States

**Competing interest:** The authors declare that no competing interests exist.

## Editor's evaluation

Sorkac et al. presents a novel genetically encoded retrograde synaptic tracing method that has the potential for unbiased identification of presynaptically connected neurons. *retro*-Tango is based on the previously developed anterograde method *trans*-Tango, promising high applicability and rendering the significance of this contribution important and for some applications fundamental. The strength of the evidence is compelling and the discussion of the technique's applicability and limitations is exceptional.

## Introduction

The Turkish poet Nazım Hikmet wrote:

> *To live, like a tree one and free*
> *And like a forest, sisterly* (*Hikmet, 2002*).

This also holds true to the function of the nervous system. Like forests, neural circuits have evolved as congruous networks of individual units: neurons. These networks integrate external stimuli with the internal state of the animal and generate the proper behavioral responses to the changing environment. Therefore, understanding the individual neuron is invaluable for deciphering animal behavior; yet the study of circuits is an indispensable complement to it.

The study of neural circuits encompasses a variety of approaches of which the analysis of connectivity between neurons is fundamental. In this respect, the complete electron microscopy (EM) reconstruction of the *Caenorhabditis elegans* nervous system in the 1980s (*White et al., 1986*) and the ongoing efforts to complete the *Drosophila melanogaster* connectome (*Bates et al., 2020b*; *Eichler et al., 2017*; *Engert et al., 2022*; *Fushiki et al., 2016*; *Horne et al., 2018*; *Hulse et al., 2021*; *Marin et al., 2020*; *Ohyama et al., 2015*; *Scheffer et al., 2020*; *Takemura et al., 2017a*; *Takemura*

*et al., 2017b*; *Zheng et al., 2018*) provide the gold standard for the analysis of neural circuits. These endeavors open new paths for the study of nervous systems. However, like all methods, they come with their own shortcomings.

The EM reconstruction of the *C. elegans* nervous system was originally performed with a single hermaphrodite reared at specific laboratory conditions. Further, it was not until 30 years later that the nervous system of a second animal, a male, was reconstructed (*Cook et al., 2019*). As to *D. melanogaster*, the brain of a single female is still being reconstructed. These time-consuming and labor-intensive aspects of EM reconstructions preclude the study of individual differences that might arise from variances such as sex, genetics, epigenetics, rearing conditions, and past experiences. Hence, transsynaptic tracing techniques remain valuable even in the age of EM connectomics.

In *D. melanogaster*, techniques such as photoactivatable GFP (PA-GFP) (*Datta et al., 2008*; *Patterson and Lippincott-Schwartz, 2002*) and GFP-reconstitution across synaptic partners (GRASP) *Fan et al., 2013*; *Feinberg et al., 2008*; *Gordon and Scott, 2009*; *Macpherson et al., 2015*; *Shearin et al., 2018* have been instrumental in studying neural circuits and connectivity. Recently, two methods, *trans*-Tango (*Talay et al., 2017*) and TRACT (*Huang et al., 2017*), were developed for anterograde transsynaptic tracing. In addition, a retrograde transsynaptic tracing method, termed BAcTrace, was devised (*Cachero et al., 2020*). All three techniques differ from the aforementioned PA-GFP and GRASP in that they provide genetic access to synaptic partners of a set of neurons, enabling their use in not only tracing but also monitoring and manipulation of neural circuits (*Snell et al., 2022*). Furthermore, *trans*-Tango and TRACT do not necessitate hypotheses prior to experimentation, since all neurons are capable of revealing the postsynaptic signal should the cascades be triggered by their presynaptic partners. In contrast, BAcTrace, by design, relies on the expression of the presynaptic components of the cascade solely in candidate neurons. Therefore, it requires a hypothesis to be tested, rendering this technique inherently biased. In addition, BAcTrace experiments are constrained by the availability of drivers in candidate neurons because the presynaptic components are expressed under a LexA driver. Hence, there is still a need for a versatile retrograde tracing method that can be used as a hypothesis tester, and, more importantly, as a hypothesis generator.

To fill this gap, here we present *retro*-Tango, a retrograde version of *trans*-Tango, as a user-friendly, versatile retrograde transsynaptic tracing technique for use in *D. melanogaster*. Like *trans*-Tango, *retro*-Tango functions through a signaling cascade initiated by a ligand-receptor interaction at the synapse and resulting in reporter expression in synaptic partners. To target the reporter expression to presynaptic neurons, we devised a ligand tethered to a protein that localizes to dendrites in the starter neurons. In order to benchmark the system, we used it in various known circuits. First, we revealed the presynaptic partners of the giant fiber from the escape circuit and compared our results to the EM reconstruction. Second, to demonstrate the versatility of *retro*-Tango, we implemented it in the central complex. Third, we tested the specificity of the system by using it in a sexually dimorphic circuit where the presynaptic partners of a set of neurons differ between males and females. Lastly, we used *retro*-Tango in the sex peptide circuit and in the olfactory system where we traced connections from the central nervous system (CNS) to the periphery and vice versa. Importantly, we compared the signal with *retro*-Tango and *trans*-Tango using the same driver and observed distinct patterns of labeling. Taken together, our experiments establish *retro*-Tango as a prime method for neuroscience research in fruit flies.

## Results

### Design of *retro*-Tango

*retro*-Tango is the retrograde counterpart of the transsynaptic tracing technique *trans*-Tango (*Talay et al., 2017*), and both are based on the Tango assay for G-protein coupled receptors (GPCRs) (*Barnea et al., 2008*). In the Tango assay, activation of a GPCR by its ligand is monitored via a signaling cascade that eventually results in reporter gene expression. This signaling cascade comprises two fusion proteins. The first is a GPCR tethered to a transcriptional activator via a cleavage site recognized by the tobacco etch virus N1a protease (TEV). The second is the human β-arrestin2 protein fused to TEV (Arr::TEV). A third component is a reporter gene under control of the transcriptional activator. Upon binding of the ligand to the receptor, arrestin is recruited to the activated receptor bringing TEV in close proximity to its recognition site. TEV-mediated cleavage then releases the transcriptional

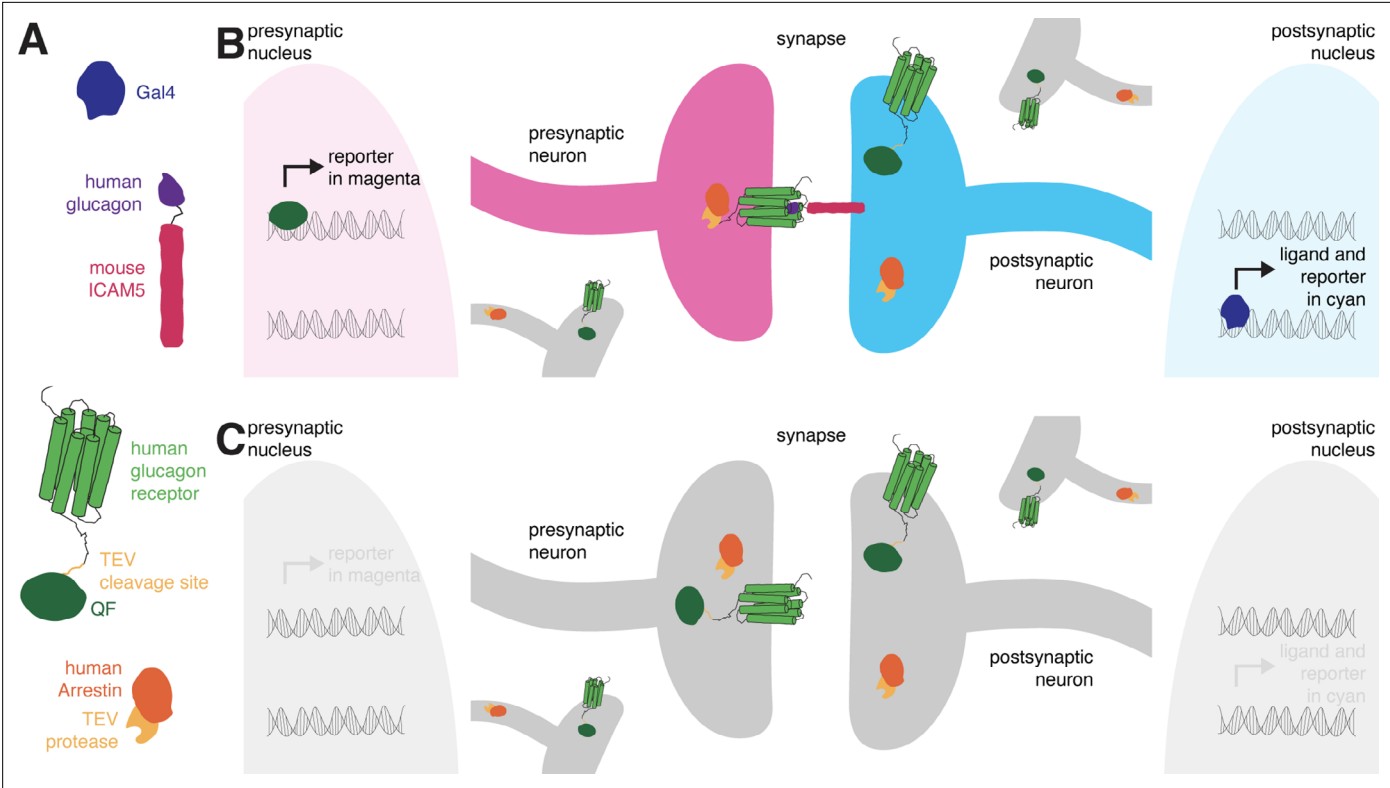

**Figure 1.** The design of *retro*-Tango. (**A**) The components of *retro*-Tango. (**B**) In *retro*-Tango, all neurons express two of the components of the signaling cascade: human glucagon receptor::TEV cleavage site::QF and human β-arrestin2::TEV protease. They also carry the gene encoding the presynaptic mtdTomato reporter (magenta) under the control of QF. Therefore, all neurons are capable of expressing the reporter. In starter neurons expressing Gal4, the ligand (human glucagon::mouse ICAM5) is expressed along with the GFP reporter (cyan) marking the postsynaptic starter neurons. The mICAM5 fusion localizes the ligand to the postsynaptic sites such that the ligand activates its receptor only across the synapse. Upon activation of the receptor in the presynaptic neuron, the Arrestin-TEV fusion is recruited. TEV-mediated proteolytic cleavage then releases the transcription factor QF from the receptor. QF in turn translocates to the nucleus and initiates transcription of the presynaptic magenta reporter. In neurons that are not presynaptic to the starter neurons, the reporter is not expressed. (**C**) In the absence of a Gal4 driver, the ligand is not expressed, and the signaling cascade is not triggered, resulting in no expression of the reporters.

The online version of this article includes the following figure supplement(s) for figure 1:

**Figure supplement 1.** The *retro*-Tango ligand localizes to dendrites and somata.

**Figure supplement 2.** The genetic components of *retro*-Tango.

activator that in turn translocates to the nucleus to initiate transcription of the reporter gene. These components are conserved in both transsynaptic tracing techniques, *trans*-Tango (*Talay et al., 2017*) and *retro*-Tango. The novelty in both methods is in the tethering of the ligand to a transmembrane protein to localize it to pre- (*trans*-Tango), or post- (*retro*-Tango) synaptic sites. In this manner, the ligand activates its receptor only across the synaptic cleft and initiates the signaling cascade in synaptic partners. In both methods, the human glucagon (GCG) and the human glucagon receptor (GCGR) are used as the ligand-receptor pair, and the GCGR is tethered to the transcriptional activator QF (GCGR::TEVcs::QF) (*Figure 1A*).

In *retro*-Tango, the targeting of glucagon to postsynaptic sites is achieved via the mouse intercellular adhesion molecule ICAM5 (*Figure 1A*). When expressed in *Drosophila* neurons, this protein is present at low levels in cell bodies and mainly localizes to the dendrites but not the axons, enabling its use as a dendritic marker (*Nicolaï et al., 2010*). Indeed, upon expression in the Kenyon cells of the mushroom body, the *retro*-Tango ligand localizes to the cell bodies and the mushroom body calyx, where the dendrites of Kenyon cells are present (*Figure 1—figure supplement 1*). By contrast, the ligand does not colocalize with Synaptotagmin1, a protein that labels presynaptic termini (*Figure 1—figure supplement 1*). In *retro*-Tango, the ligand and the postsynaptic reporter farnesylated GFP are stoichiometrically expressed under the control of the Gal4/UAS system via the self-cleaving

P2A peptide (*Daniels et al., 2014*; *Figure 1—figure supplement 2*). In this manner, the presence of the ligand is coupled with the GFP signal, eliminating any discrepancy that might arise from differentially expressing them from two separate genomic sites. Both the GCGR::TEVcs::QF and the Arr::TEV fusion proteins are expressed panneuronally, and the expression of the presynaptic reporter mtdTomato is controlled by the QF/QUAS binary system (*Potter et al., 2010*; *Figure 1—figure supplement 2*). In postsynaptic starter cells, Gal4 drives the expression of both GFP and the ligand (*Figure 1B*). The interaction of the ligand with its receptor on the presynaptic partners triggers the *retro*-Tango cascade that culminates in mtdTomato expression in these neurons. By contrast, the ligand is not expressed in the absence of a Gal4 driver. Therefore, the cascade is not triggered, and no presynaptic signal is observed (*Figure 1C*). Since the presynaptic components of the pathway are expressed panneuronally, all neurons have the capacity to reveal the presynaptic signal when the ligand is expressed by their postsynaptic partners. Thus, the design of *retro*-Tango is not inherently biased.

## Validation of *retro*-Tango

For the initial validation of *retro*-Tango, we chose the giant fibers (GFs) of the escape circuit. The GFs are descending command interneurons that respond to neural pathways sensing looming stimuli, such as from a predator. They then relay this information to downstream neurons for the fly to initiate the take-off response (*Fotowat et al., 2009*; *von Reyn et al., 2014*). The GFs receive direct input from two types of visual projection neurons: lobula columnar type 4 (LC4) (*von Reyn et al., 2017*) and lobula plate/lobula columnar type 2 (LPLC2) (*Ache et al., 2019*). They then integrate this information and convey it to the tergotrochanteral motor neurons (TTMns) and the peripherally synapsing inter-neurons (PSIs) in the ventral nerve cord (VNC). The GFs form chemical and electrical synapses with both of these types of neurons (*Allen et al., 2006*). All of these neurons are easily identifiable based on their morphology in the optic lobes or the VNC, rendering the GF system attractive for validating *retro*-Tango. In addition, there is a specific driver line that expresses only in the GFs (*von Reyn et al., 2014*). Further, the GFs are clearly annotated in the EM reconstruction of the hemibrain (*Zheng et al., 2018*), allowing for the comparison of the *retro*-Tango results with the annotated connectome.

When we initiated *retro*-Tango from the GFs in adult males, we observed strong presynaptic signal in cells with dense arborizations in the brain and sparse processes in the VNC (*Figure 2A*). Upon close examination, we noticed few cell bodies in the VNC, suggesting that the VNC signal originates mostly from descending neurons with somata in the brain. As expected, we did not observe *retro*-Tango signal in the TTMns and PSIs, known postsynaptic partners of the GFs. Importantly, we could identify neurons in the optic lobes with the characteristic dendritic arborizations of the LC4s and the LPLC2s, established presynaptic partners of the GFs. By contrast, when we initiated *trans*-Tango from the GFs, we observed labeling in their predicted postsynaptic partners (*Figure 2—figure supplement 1A*). In addition, in *trans*-Tango experiments, there was little to no signal in the brain. Together, these results show that *retro*-Tango does not work in the anterograde direction. It is noteworthy that in *retro*-Tango we observed sporadic asymmetrical signal in the postsynaptic starter neurons, a phenom-enon we notice when we use some split-Gal4 drivers. Likewise, we observe asymmetry in the *retro*-Tango signal in the presynaptic neurons. The stronger signals in the postsynaptic and the presynaptic neurons are in the same hemisphere, likely reflecting higher ligand expression in the starter neurons. Such differences in signal intensity may lead to qualitative differences in presynaptic neurons revealed in each hemisphere. For example, the LC4 neurons (marked by the arrow) are visible only in one hemisphere (*Figure 2A*). Nonetheless, we conclude that *retro*-Tango yields strong signal and labels the expected presynaptic partners of the GFs. Further, it does not exhibit false positive signal in the postsynaptic targets of the GFs. These results indicate that *retro*-Tango is indeed selective to the retrograde direction.

It is noteworthy that we do not observe strong background noise with *retro*-Tango in the absence of a Gal4 driver where the ligand is not expressed (*Figure 2B*). There is, however, faint background noise in some of the Kenyon cells of the mushroom body as well as in the fan-shaped body and noduli of the central complex. In addition, we occasionally observe sporadic noise in a few neurons in the VNC. This background noise might be due to leaky expression of the ligand, albeit in low levels as reflected by the absence of the GFP signal. Alternatively, it might be due to leaky expression of the postsynaptic reporter mtdTomato itself.

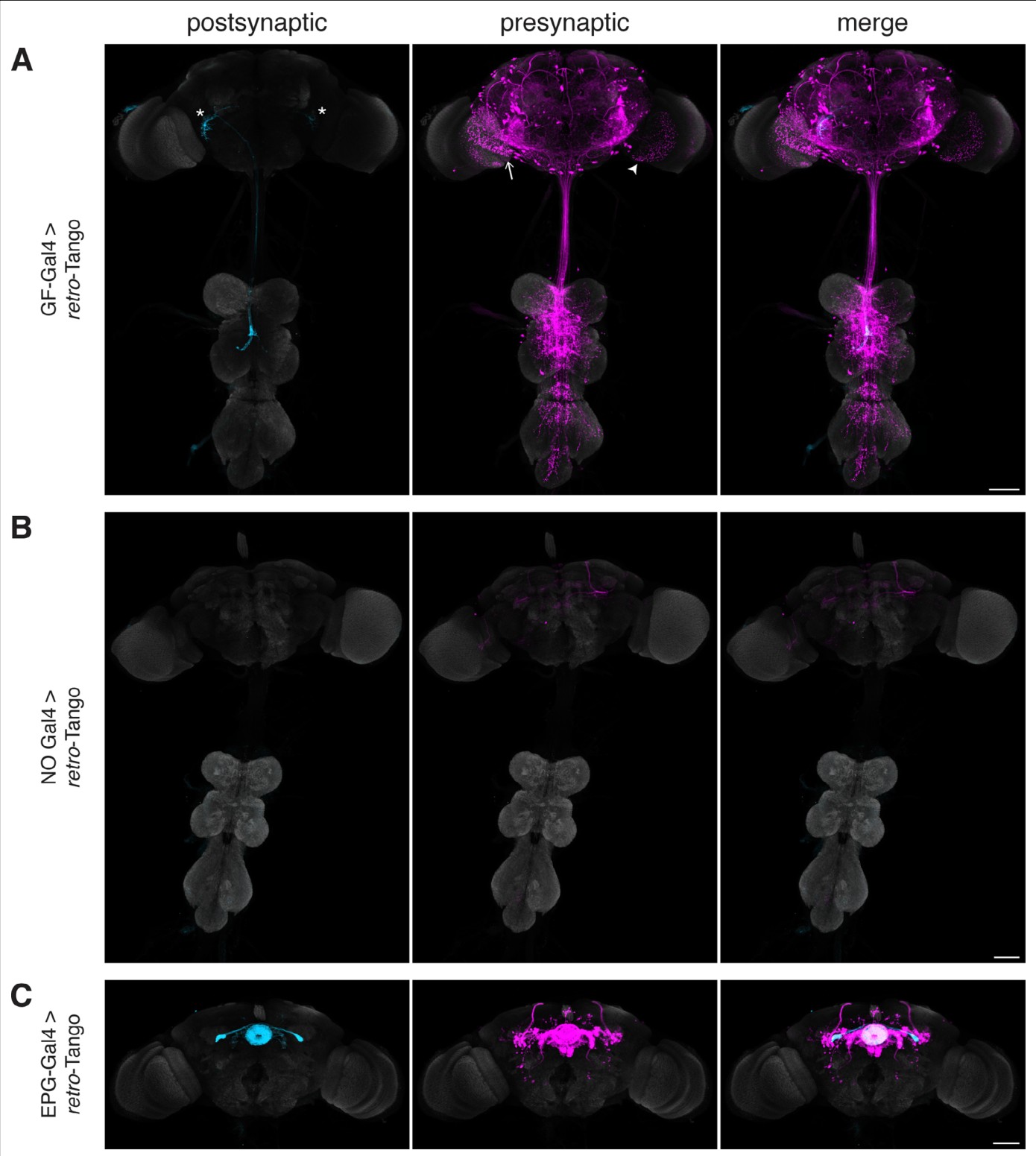

**Figure 2.** Implementation of *retro*-Tango in the giant fiber and central complex circuits. (**A**) Initiating *retro*-Tango from the GFs (asterisks mark the cell bodies) results in presynaptic signal in the brain and VNC (223±59 neurons in 5 brains, 1±3 neurons in 5 VNCs). Both LC4 (arrow) and LPLC2 (arrowhead) neurons, known presynaptic partners of GFs, are identified by *retro*-Tango. Note the asymmetry between hemispheres in the signal in the postsynaptic starter neurons and their corresponding presynaptic partners. (**B**) *retro*-Tango exhibits little background noise in the absence of a Gal4 driver. Background is observed in the mushroom bodies, in the central complex, and in a few neurons in the VNC (68±10 neurons in 4 brains, 1±1 neurons in 4 VNCs). (**C**) Ligand expression in EPG neurons of the central complex leads to *retro*-Tango signal in their known presynaptic partners: PEN, PFR and Δ7

*Figure 2 continued on next page*

*Figure 2 continued*

neurons (170±24 neurons in 5 brains). The signal in these neurons can be easily discerned from the background noise. 15do males were analyzed for all panels. Postsynaptic GFP (cyan), presynaptic mtdTomato (magenta) and neuropil (grey). Scale bars, 50 μm.

The online version of this article includes the following video and figure supplement(s) for figure 2:

**Figure supplement 1.** *trans*-Tango in the giant fiber and central complex circuits.

**Figure supplement 2.** The *retro*-Tango signal in the EPG circuit is far stronger than the background noise.

**Figure supplement 3.** *retro*-Tango does not yield false positive signal in neighboring neurons in the EPG circuit.

**Figure 2—video 1.** *retro*-Tango does not yield false positive signal in neighboring neurons in the EPG circuit.

https://elifesciences.org/articles/85041/figures#fig2video1

In view of the faint background noise that we observed in some brain regions, we decided to examine whether *retro*-Tango can be used in one of these regions, the central complex.

The central complex is a series of interconnected neuropil structures that are thought to act as the major navigation center of the fly brain. The flow of information through the central complex indicates that it dynamically integrates various sensory cues with the animal's internal state for goal-directed locomotion (*Hulse et al., 2021*). In the central complex circuitry, ellipsoid body-protocerebral bridge-gall (EPG) neurons have dendrites in the ellipsoid body (EB) and axons in the protocerebral bridge (PB) as well as in the lateral accessory lobes (LALs). EPGs are the postsynaptic targets of the ring neurons of the EB. They also form reciprocal connections with PB-EB-noduli (PEN) neurons, PB-fan shaped body-round body (PFR) neurons and Δ7 interneurons (*Hulse et al., 2021*; *Seelig and Jayaraman, 2013*; *Sun et al., 2017*). When we initiated *retro*-Tango from the EPGs, we observed presynaptic signal in the predicted presynaptic partners (*Figure 2C*).

In light of the known reciprocal connections in the central complex, we sought to examine whether initiating *retro*-Tango and *trans*-Tango from the same population of neurons would result in differential labeling. Indeed, driving *trans*-Tango from the EPGs revealed an overlapping yet different pattern than *retro*-Tango (*Figure 2—figure supplement 1B*). Since the ring neurons of the EB are solely presynaptic to the EPGs, the *trans*-Tango-mediated postsynaptic signal in the EB is far weaker than the presynaptic *retro*-Tango signal (*Figure 2C*, *Figure 2—figure supplement 1B and C*). By contrast, *trans*-Tango reveals strong signal in the LALs where the axons of the EPGs meet the dendrites of their postsynaptic partners, while there is virtually no presynaptic signal in the LALs with *retro*-Tango (*Figure 2C*, *Figure 2—figure supplement 1B and D*). These results further indicate that *retro*-Tango exclusively functions in the retrograde direction.

Importantly, initiating *retro*-Tango from the EPGs resulted in a much stronger signal in the central complex than the noise we observed in the absence of a driver (*Figure 2—figure supplement 2*). This observation indicates that *retro*-Tango can indeed be used in brain regions with background noise.

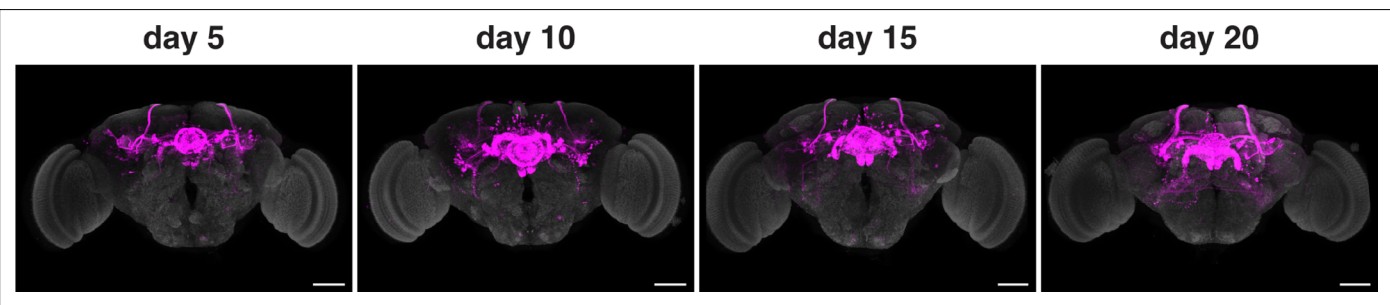

**Figure 3.** Age dependence of *retro*-Tango. The *retro*-Tango signal is observed in 5 day intervals upon ligand expression in the EPGs. The signal accumulates with time and saturates around day 10 post-eclosion. Males were analyzed for all panels. Presynaptic mtdTomato (magenta) and neuropil (grey). Scale bars, 50 μm.

The online version of this article includes the following figure supplement(s) for figure 3:

**Figure supplement 1.** Quantification of the pixel intensity for the signal of *retro*-Tango when initiated from the EPG neurons.

**Figure supplement 2.** Age dependence of the *retro*-Tango signal in the presynaptic partners of the GFs in males.

**Figure supplement 3.** Age dependence of the *retro*-Tango signal in the presynaptic partners of the GFs in females heterozygous for the reporter.

Further, the absence of labeling in any unexpected neuronal processes near the EPG cell bodies suggests that *retro*-Tango does not lead to false positive signal due to the presence of its ligand in neuronal somata (*Figure 2—figure supplement 3*, *Figure 2—video 1*). Finally, we do not observe presynaptic signal in starter neurons, indicating that expression of the *retro*-Tango ligand in a starter neuron does not activate the signaling pathway in the same cell (*Figure 2—figure supplement 3*, *Figure 2—video 1*).

We next sought to test the age-dependence of the presynaptic signal in *retro*-Tango. We initiated *retro*-Tango from the EPGs and examined the signal in adults at days 5, 10, 15, and 20 post-eclosion (*Figure 3*). We noticed that the signal accumulates and reaches saturation around day 10 post-eclosion (*Figure 3—figure supplement 1*). A similar analysis with GFs as the starter neurons indicated that the signal keeps accumulating over time in males (*Figure 3—figure supplement 2*) but not in females heterozygous for the reporter (*Figure 3—figure supplement 3*). Therefore, we concluded that the accumulation of the *retro*-Tango signal depends on the circuit of interest, and possibly, on the strength of the driver line being used. To be prudent, we examined adult flies 15 days post-eclosion for the remainder of the study.

## Comparison of *retro*-Tango with the EM reconstruction of the female hemibrain

Having established the system in the GF and EPG circuits, we wished to benchmark it by comparing the presynaptic signal of *retro*-Tango with the EM reconstruction of the female hemibrain. In the connectome, we found 1101 neurons presynaptic to the giant fiber (*Figure 4—figure supplement 1A*). We observed fewer (223±60 neurons in 5 brains) presynaptic neurons with *retro*-Tango (*Figure 2A*). Based on the EM reconstruction, the number of synapses that these 1101 neurons form with the GF ranges from 1 to 380. We, therefore, reasoned that the number of synapses that a given presynaptic neuron forms with the starter neuron affects whether it is labeled by *retro*-Tango. In other words, there is a threshold in the number of synapses that a presynaptic neuron makes with a starter neuron under which it cannot be labeled with *retro*-Tango. Neurons with fewer synapses than this threshold likely constitute the false negatives of *retro*-Tango. This threshold could be affected by the circuit of interest and by the strength of the driver line.

To determine this threshold, we decided to count the presynaptic neurons of the GF revealed by *retro*-Tango using a nuclear reporter. In these experiments, we counted the neurons in each half of the

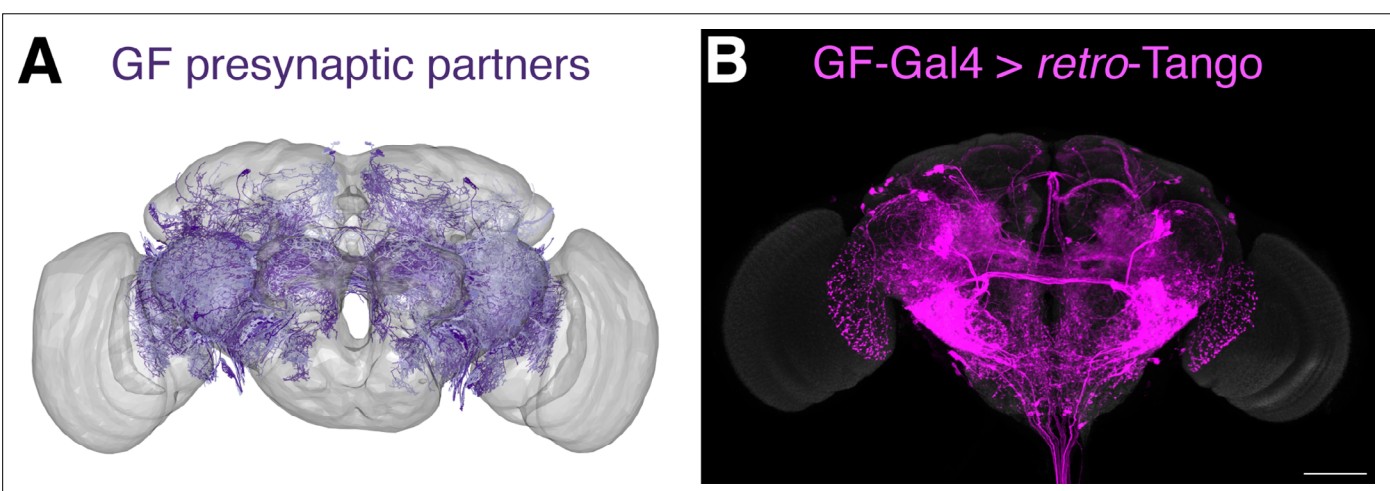

**Figure 4.** Comparison of the *retro*-Tango signal with the EM reconstruction of the female hemibrain. (**A**) Plotting of the skeletonizations of the EM segmentations of presynaptic partners that connect with the GF via 17 synapses or more. (**B**) Presynaptic partners of the GFs in a female fly as revealed by *retro*-Tango. 15do females heterozygous for the tdTomato reporter were analyzed for panel (**B**). Presynaptic mtdTomato (magenta) and neuropil (grey). Scale bar, 50 µm. Note the high similarity between the patterns in both panels.

The online version of this article includes the following figure supplement(s) for figure 4:

**Figure supplement 1.** Methodology for the comparison of *retro*-Tango results with the hemibrain connectome.

**Figure supplement 2.** *retro*-Tango reveals LPLC2s as presynaptic partners of the GF in females when the reporter is homozygous.

brain focusing on the area that is covered by the connectome (*Figure 4—figure supplement 1B and C*). We counted five experimental GF *retro*-Tango brains and observed an average of 191±31 neurons in this area. In six control brains from flies not carrying Gal4, we counted an average of 26±9 neurons. We concluded that in this area, *retro*-Tango correctly labels approximately 165 neurons when initiated from the GF. Of the 1101 neurons that the connectome reveals as presynaptic to the GF, 341 have cell bodies in the area covered by the EM reconstruction. Therefore, *retro*-Tango identifies approximately half of these neurons. We analyzed the connectome data for these 341 neurons and found that 168 of them have each 17 synapses or more with the GF. Given that *retro*-Tango reveals approximately 165 neurons, we concluded that the threshold for *retro*-Tango to identify the presynaptic partners of the GF is 17 synapses in females heterozygous for the nuclear reporter (*Figure 4—figure supplement 1A*).

We subsequently used this newly determined threshold to sort the 1101 neurons revealed by the connectome as presynaptic to the GF and identified 265 neurons. We then plotted the skeletonizations of the EM segmentations of these 265 neurons (*Figure 4A*). When we initiated *retro*-Tango from the GF in females heterozygous for the reporter, we revealed a strikingly similar pattern (*Figure 4B*). It is noteworthy that we observe some differences in the *retro*-Tango signal between males and females. Based on the connectome, LPLC2s form an average of 13 synapses per neuron with the giant fiber (*Ache et al., 2019*). This is below the threshold, and indeed, we do not observe LPLC2s in females heterozygous for the *retro*-Tango reporter (*Figure 4B*). By contrast, we do observe them in males (*Figure 2A*). This discrepancy could be explained by the location of the presynaptic mtdTomato reporter on the X-chromosome. Accordingly, the reporter expression level in males is higher compared to heterozygous females due to X-chromosome upregulation for dosage compensation (*Gorchakov et al., 2009*). To test this, we analyzed females homozygous for the presynaptic reporter. In these animals, *retro*-Tango revealed the LPLC2s as presynaptic to the GFs (*Figure 4—figure supplement 2*) indicating that doubling of the reporter on the X-chromosome increases the sensitivity of *retro*-Tango. Thus, the threshold for *retro*-Tango to reveal the presynaptic partners in hemizygous males or homozygous females is significantly lower than in heterozygous females. This threshold also depends on the age at which the animals are dissected since the *retro*-Tango signal may accumulate with age (*Figure 3—figure supplement 2*).

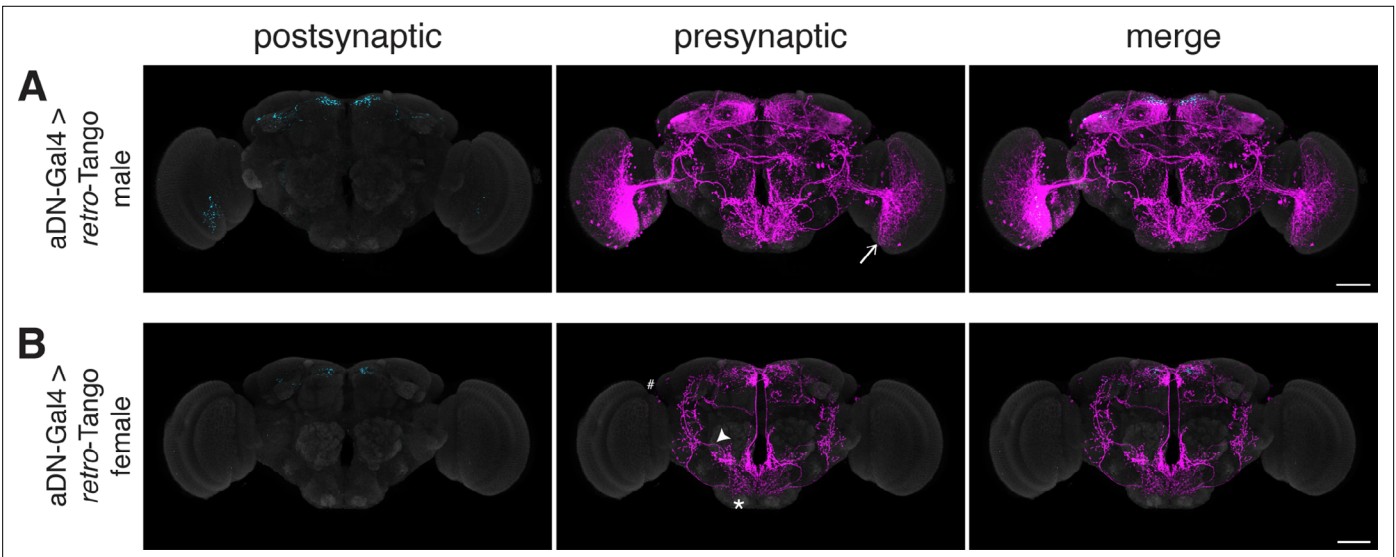

**Figure 5.** Assessing the specificity of *retro*-Tango in a sexually dimorphic circuit. (**A**) Initiating *retro*-Tango in aDNs in male flies reveals visual projection neurons (arrow) as presynaptic partners (223±59 neurons in 5 brains). (**B**) Initiating *retro*-Tango in aDNs in females results in presynaptic reporter expression in the lateral antennal lobe tract (arrowhead), the SEZ (asterisk), and the LH (hash) (24±11 neurons in 5 brains). 15do males hemizygous for the tdTomato reporter (**A**) and females heterozygous for the reporter (**B**) were analyzed. Postsynaptic GFP (cyan), presynaptic mtdTomato (magenta) and neuropil (grey). Scale bars, 50 μm.

## Specificity of *retro*-Tango

Having benchmarked *retro*-Tango in tracing various connections, we sought to determine its specificity and reasoned that sexually dimorphic circuits would be apposite for this analysis. One such circuit involves the anterior dorsal neurons (aDNs), a pair of neurons in each hemisphere that receive inputs from distinct sensory systems in the two sexes. In males, the aDNs receive visual input, whereas in females, the input instead comes from the olfactory and thermo/hygrosensory systems (*Nojima et al., 2021*). Thus, we decided to use the sexual dimorphism in the inputs to aDNs for testing the specificity of *retro*-Tango. When we initiated *retro*-Tango from aDNs in males, we observed strong presynaptic signal in the central brain, and more importantly, in the visual system (*Figure 5a*). However, we did not observe presynaptic signal in LC10 neurons as would be predicted (*Nojima et al., 2021*). A possible explanation for the absence of labeling in LC10s could be that the strength of connections between LC10s and aDNs is below the detection threshold of *retro*-Tango. Alternatively, LC10s may not be directly presynaptic to aDNs as the connections between these neurons were revealed by a non-synaptic version of GRASP (*Gordon and Scott, 2009*; *Nojima et al., 2021*). By contrast, in females, we observed two neurons in the lateral antennal lobe tracts, few neurons in the lateral horns (LHs), and neuronal processes in the suboesophageal zone (SEZ) as previously reported (*Figure 5B*). However, the signal in females is low, likely because they are heterozygous for the presynaptic reporter. Indeed, it seems that *retro*-Tango does not identify all the presynaptic neurons reported in females (*Nojima et al., 2021*). Nonetheless, the difference in the signal pattern between male and female brains demonstrates the specificity of *retro*-Tango.

## Using *retro*-Tango to trace connections between the CNS and the periphery

Our experiments in the giant fiber, the central complex circuits and the aDNs established *retro*-Tango for tracing connections within the CNS. Next, we wished to examine whether *retro*-Tango can be used to trace connections between the CNS and the periphery. To achieve this, we turned to two well-characterized circuits: the sex peptide (SP) circuit and the olfactory circuit.

The SP circuit mediates the response of females to the presence of SP in the seminal fluid upon mating. SP is detected by the SP sensory neurons (SPSNs) located in the lower reproductive tract of females (*Yapici et al., 2008*). SPSNs project to the SP abdominal ganglion (SAG) neurons in the CNS to initiate the post-mating switch, a set of programs that alter the internal state of the female (*Feng et al., 2014*). Accordingly, initiating *retro*-Tango from SAG neurons reveals presynaptic signal in a pair of neurons in the lower reproductive tract, consistent with SPSNs (*Figure 6A*). This result confirms that *retro*-Tango can be used to reveal connections between the CNS and the periphery.

In the olfactory circuit, olfactory receptor neurons (ORNs) located in the antennae and the maxillary palps, the two olfactory sensory organs, project their axons to the antennal lobe, a brain region consisting of multiple neuropil structures called glomeruli. The ORNs that express the same olfactory receptor converge on the same glomerulus where they form synapses with lateral interneurons (LNs) and olfactory projection neurons (OPNs). The OPNs, in turn, relay the information to higher brain areas, primarily the mushroom body (MB) and the LH. Thus, in a simplistic model, the flow of sensory information is from the ORNs to the OPNs while LNs form synapses with both neuronal types. However, all three neuronal types are interconnected via reciprocal synapses (*Horne et al., 2018*). Therefore, in this circuit, if we initiate *retro*-Tango in the ORNs, we expect to see presynaptic signal in the OPNs and LNs. We, hence, sought to test *retro*-Tango in these reciprocal synapses. To this end, we initiated *retro*-Tango from a subset of ORNs that express the olfactory receptor Or67d and project to the DA1 glomeruli. We, indeed, observed presynaptic signal in OPNs and LNs (*Figure 6B*). By contrast, when we initiated *trans*-Tango from the same neurons, we revealed a much stronger signal with some distinct patterns (*Figure 6—figure supplement 1*). For instance, the mediolateral antennal lobe tract, clearly visible with *trans*-Tango, is absent in *retro*-Tango. The distinction between the signals with the two systems can be explained by the higher number of synapses where ORNs are presynaptic to OPNs and LNs than vice versa (*Horne et al., 2018*). Further, the dissimilarity in the signal patterns observed with *retro*-Tango and *trans*-Tango demonstrates the absence of the *retro*-Tango ligand from the presynaptic sites. Together, these results confirm that *retro*-Tango can be used to reveal synaptic connections between the CNS and the periphery irrespective of the direction of information flow.

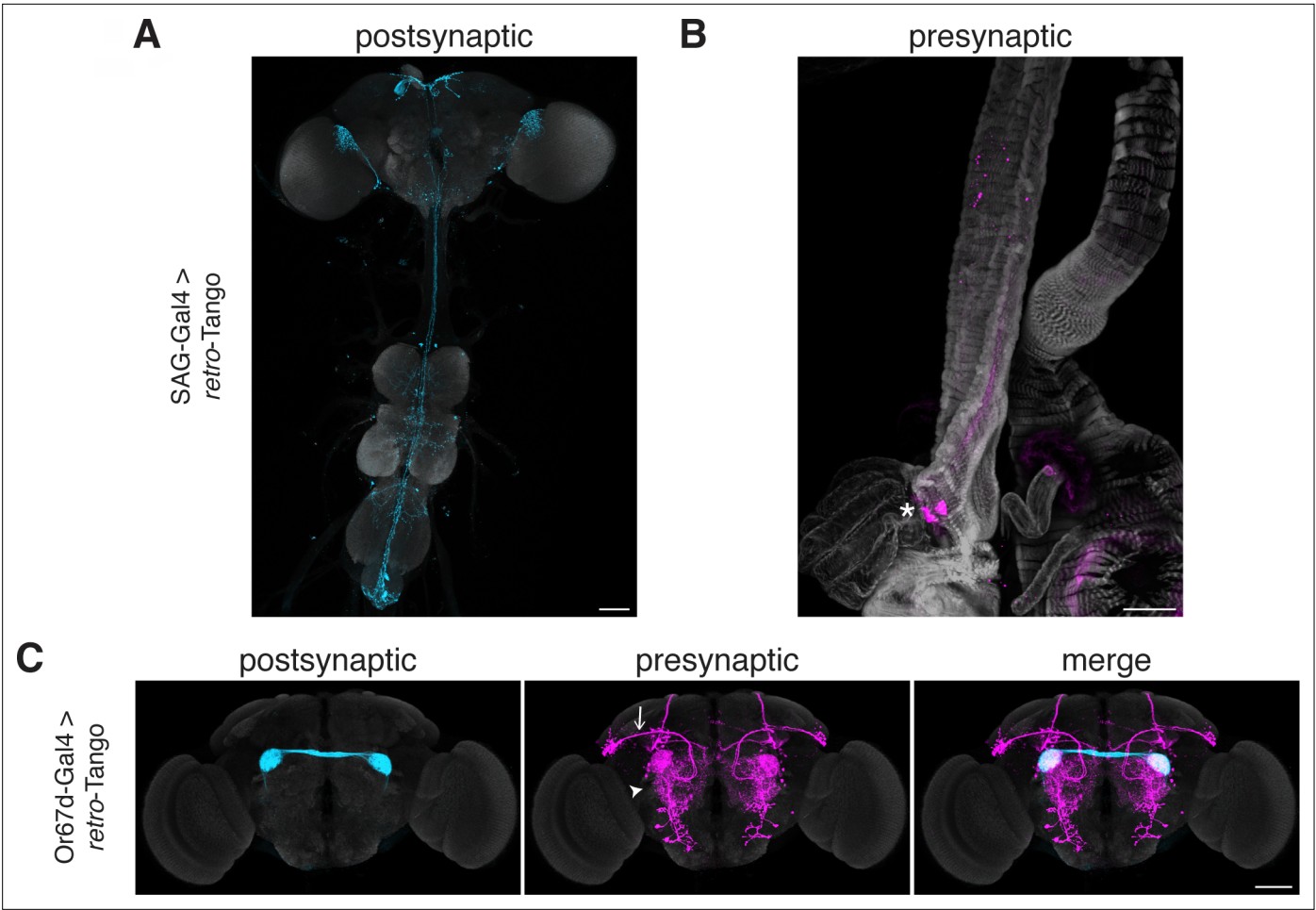

**Figure 6.** Tracing connections between the periphery and the CNS with *retro*-Tango. (**A**) Expression of the *retro*-Tango ligand in SAG neurons reveals (**B**) SPSNs (asterisk) as presynaptic partners. (**C**) When *retro*-Tango is initiated from Or67d-expressing ORNs, OPNs (arrow) and LNs (arrowhead) are revealed as their presynaptic partners(134±17 neurons in 5 brains). 15do females heterozygous for the tdTomato reporter (**A**) and males (**B**) were analyzed. Postsynaptic GFP (cyan), presynaptic mtdTomato (magenta) and neuropil (**A, C**), or phalloidin (**B**) (grey). Scale bars, 50 μm.

The online version of this article includes the following figure supplement(s) for figure 6:

**Figure supplement 1.** Initiating *trans*-Tango from the Or67d-expressing ORNs.

## Discussion

In this study, we presented *retro*-Tango, a new method for retrograde transsynaptic tracing in *Drosophila*. *retro*-Tango is a versatile retrograde tracing method that can be used both as a hypothesis tester and a hypothesis generator. It shares many of its components with *trans*-Tango (**Talay et al., 2017**) and differs from it in the transmembrane protein with which the ligand is delivered. In *trans*-Tango a dNeurexin1-hICAM1 chimeric protein localizes the ligand to presynaptic sites such that it activates its receptor only in postsynaptic neurons across the synaptic cleft (**Talay et al., 2017**). By contrast, in *retro*-Tango the ligand is attached to mICAM5, a dendritic marker in *Drosophila* (**Nicolaï et al., 2010**). Thus, driving the *retro*-Tango ligand in starter neurons activates the receptor in their presynaptic partners. This, in turn, triggers the signaling cascade culminating in reporter gene expression in the presynaptic neurons.

We used the GF circuit to validate *retro*-Tango since some of the known synaptic partners of the GFs can be easily identified. These experiments confirmed that *retro*-Tango correctly labels the expected presynaptic partners. In addition, we did not observe signal in the postsynaptic partners of the GFs, indicating that *retro*-Tango does not falsely label in an anterograde fashion. Further, driving ligand expression results in strong signal in the presynaptic neurons, while without a driver, the background noise is weak. We observed noise mainly in the MBs and the central complex with sporadic labeling in

the VNC. To assess the utility of *retro*-Tango in these areas, we implemented it in the central complex. These experiments revealed presynaptic signal that can easily be discerned from the noise. That said, users should be cautious in drawing strong conclusions from *retro*-Tango experiments in these areas. As in *trans*-Tango (*Talay et al., 2017*), the panneuronal components are inserted at the attP40 docking site in the genome. It is noteworthy that the attP40 docking site has recently been shown to cause problems in the nervous system, especially when homozygous (*Duan et al., 2023*; *Groen et al., 2022*; *van der Graaf et al., 2022*). Therefore, we advise against using the panneuronal components in a homozygous configuration. Likewise, users should be cautious when using Gal4 or split Gal4 lines inserted at the attP40 site.

The expression of mICAM5 is not entirely restricted to dendrites. Rather, it is also expressed in the somata, albeit at low levels (*Nicolaï et al., 2010*). Hence, we were concerned that this would lead to labeling in neighboring neurons that are not true synaptic partners. However, our experiments in the central complex indicated that this is not the case. Nevertheless, caution should be taken especially when using strong drivers. It is also worth mentioning that we do not observe presynaptic labeling in the starter neurons, indicating that *retro*-Tango only works between cells.

Unlike *trans*-Tango (*Talay et al., 2017*), *retro*-Tango yields strong signal at 25°C. This feature of *retro*-Tango is especially important as a recent study showed that the number of synaptic partners of a neuron and the number of connections with each partner are inversely correlated with rearing temperature (*Kiral et al., 2021*). Therefore, using *retro*-Tango at 25°C prevents inconsistencies with other experiments run at this temperature. In addition, while like in *trans*-Tango (*Talay et al., 2017*) the signal in *retro*-Tango correlates with age, it accumulates faster. Although in some circuits, such as the GF, the signal keeps increasing over time, in others, such as the EPG, it saturates by day 10 post-eclosion. The difference in saturation times could be due to the strength of the drivers or reflect the specific characteristics of the circuits. Therefore, users should determine the optimal age for analysis depending on the circuit studied and driver used.

The availability of the annotated connectome data for the female hemibrain (*Zheng et al., 2018*) enabled us to benchmark the results obtained with *retro*-Tango and assess its sensitivity. To this end, we compared our results in the GF circuit to the annotated female hemibrain connectome (*Zheng et al., 2018*; *Figure 4*). Our initial analysis indicated that *retro*-Tango falls short of revealing all the GF synaptic partners predicted by the connectome. Notably, some of these partners form single or few synapses with the GF. Therefore, it is possible that *retro*-Tango is not sensitive enough to reveal these weak connections. In our comparison, we determined the threshold for the number of synapses required for *retro*-Tango to correctly reveal a connection in the GF circuit in females heterozygous for the nuclear reporter. We applied this threshold to sort the presynaptic partners of the GF in the hemibrain connectome. When we plotted the neurons forming more synapses than the threshold, we observed a similar pattern to that revealed by *retro*-Tango. However, albeit useful for giving a general estimate about the false negatives of *retro*-Tango, this approach has certain shortcomings. The likelihood that *retro*-Tango would reveal a presynaptic partner does not rely solely on the number of synapses but also on their strength. Moreover, we found that this threshold depends on the zygosity of the reporter on the X-chromosome and therefore, on the sex of the animal. In addition, the threshold we determined only applies to the GF circuit with the specific driver we used. This threshold is bound to be different in other neural circuits. Even within the same circuit, the nature of the reporter protein, and the level of expression for the *retro*-Tango ligand will likely affect it, with strong drivers resulting in lower threshold values. Finally, it is conceivable that stochastic events at every level of the system may play a role in *retro*-Tango labeling. Hence, the value of the threshold that we determined should only be used as a general estimate, rather than an absolute value that reflects the performance of *retro*-Tango in every circuit.

Although *retro*-Tango can be used to reveal connections in most circuits, there may be instances where it does not yield useful results. For instance, when initiated from the OPNs, *retro*-Tango falls short of labeling the ORNs. This may be due to the strength of the driver (GH146) used to initiate *retro*-Tango, or it may reflect an intrinsic bias of the system against these connections. In addition, *retro*-Tango from Kenyon cells reveals signal in so many neurons that the analysis of the presynaptic partners is extremely difficult. In instances like this, *retro*-Tango can be coupled with mosaic analysis such as MARCM (*Lee and Luo, 1999*) or Flp-out (*Gordon and Scott, 2009*) to reveal a subset of the presynaptic partners. Alternatively, BAcTrace (*Cachero et al., 2020*) may be used to overcome this

problem. Finally, in the Or67d circuit, we attribute the similarity between the *retro*-Tango and *trans*-Tango signals to the known reciprocal connections between ORNs, OPNs and LNs (**Horne et al., 2018**). The clear distinction between the signals in the other two circuits (EPG and GF) supports this interpretation. That said, it is not inconceivable that with certain drivers in certain circuits some false positive signal might be observed in the anterograde direction if the ligand localizes outside the postsynaptic membrane. However, even if the *retro*-Tango ligand is only enriched in the postsynaptic membrane and not exclusively targeted there, one would expect the levels of the ligand at the presynaptic sites to be minimal and mostly below the threshold to activate the Tango cascade. Nonetheless, users should be cognizant of the possibility of anterograde labeling.

One of the features that *retro*-Tango shares with *trans*-Tango is its modular design. In *retro*-Tango, this design provides genetic access to the presynaptic neurons. Therefore, the reporter can be readily swapped with an effector that allows for monitoring (**Snell et al., 2022**), activation, or inhibition of the presynaptic neurons. In addition, like *trans*-Tango (**Coomer et al., 2023**), the modular design facilitates the adaptation of *retro*-Tango to other organisms. Notably, since using *retro*-Tango does not rely on a prior hypothesis regarding the identity of the presynaptic partners; it is flexible and general, and it can be used as a hypothesis generator. Presynaptic partners identified via *retro*-Tango can then be verified using orthogonal techniques. Thus, *retro*-Tango is a significant addition to the toolkit for studying neural circuits that can open new avenues for circuit analyses.

# Materials and methods

**Key resources table**

| Reagent type (species) or resource | Designation | Source or reference | Identifiers | Additional information |
|---|---|---|---|---|
| Genetic Reagent (*D. melanogaster*) | GF-split-Gal4 | *von Reyn et al., 2014* | RRID: BDSC#79602 | Flybase symbols: P{R17A04-p65.AD} P{R68A06-GAL4.DBD} |
| Genetic Reagent (*D. melanogaster*) | Or67d$^{Gal4}$ | *Kurtovic et al., 2007* | FlyBase: FBti0168583 | Flybase symbol: TI{GAL4}Or67d$^{GAL4-1}$ |
| Genetic Reagent (*D. melanogaster*) | ss00090-Gal4 | *Wolff and Rubin, 2018* | RRID: BDSC#75849 | Flybase symbols: P{R15C03-GAL4.DBD} P{R19G02-p65.AD} |
| Genetic Reagent (*D. melanogaster*) | SAG-split-Gal4 | *Feng et al., 2014* | RRID: BDSC#66875 | Flybase symbols: P{VT007068-GAL4.DBD} P{VT050405-p65.AD} |
| Genetic Reagent (*D. melanogaster*) | aDN-split-Gal4 | *Nojima et al., 2021* | FlyBase:FBal0243326 FlyBase: FBal0325783 | Flybase symbols: P{dVP16AD}VGlut$^{OK371-dVP16AD}$ TI{GAL4(DBD)::Zip-}dsx$^{GAL4-DBD}$ |
| Genetic Reagent (*D. melanogaster*) | QUAS-nls-DsRed | *Snell et al., 2022* | RRID: BDSC#95315 | Isolated from BDSC#95315 Flybase symbol: P{5xQUAS-nlsDsRedT4}su(Hw)attP8 |
| Genetic Reagent (*D. melanogaster*) | QUAS-mtdTomato(3xHA) | This study | | Will be deposited to Bloomington *Drosophila* Stock Center |
| Genetic Reagent (*D. melanogaster*) | *retro*-Tango(panneuronal) | This study | | Will be deposited to Bloomington *Drosophila* Stock Center |
| Genetic Reagent (*D. melanogaster*) | *retro*-Tango(ligand) | This study | | Will be deposited to Bloomington *Drosophila* Stock Center |
| Genetic Reagent (*D. melanogaster*) | MB247-Gal4 | *Aso et al., 2009* | RRID: BDSC#50742 | Flybase symbol: P{Mef2-GAL4.247} |
| Genetic Reagent (*D. melanogaster*) | UAS-syt::GFP | *Zhang et al., 2002* | RRID: BDSC#6924 | Flybase symbol: P{UAS-syt.eGFP} |
| Genetic Reagent (*D. melanogaster*) | Reporters +*trans*-Tango | *Talay et al., 2017* | RRID: BDSC#77124 | Flybase symbols: P{trans-Tango} P{UAS-myrGFP.QUAS-mtdTomato-3xHA} |

*Continued on next page*

*Continued*

| Reagent type (species) or resource | Designation | Source or reference | Identifiers | Additional information |
|---|---|---|---|---|
| Antibody | α-GFP (chicken polyclonal) | Gift from Susan Brenner-Morton (Columbia University) | | IHC (1:10000) |
| Antibody | α-RFP (guinea pig polyclonal) | Gift from Susan Brenner-Morton (Columbia University) | | IHC (1:10000) |
| Antibody | α-Brp (mouse monoclonal) | DSHB | RRID: AB_2314866 | IHC (1:20) |
| Antibody | α-chicken 488 (donkey polyclonal) | Jackson ImmunoResearch # 703-546-155 | RRID: AB_2340376 | IHC (1:1000) |
| Antibody | α-guinea pig 555 (donkey polyclonal) | Jackson ImmunoResearch # 706-165-148 | RRID: AB_2340460 | IHC (1:1000) |
| Antibody | α-mouse 647 (donkey polyclonal) | Thermo Fisher Scientific #A-31571 | RRID: AB_162542 | IHC (1:1000) |
| Chemical compound, drug | Phalloidin 647 | Thermo Fisher Scientific | Catalog number: A22287 | (1:500) |

## Fly strains

All fly lines were maintained in humidity-controlled incubators under standard 12 hr light/12 hr dark cycle. For *trans*-Tango experiments, flies were kept at 18°C; for all other experiments at 25°C. Flies were reared on standard cornmeal/agar/molasses media.

## Generation of transgenic fly lines

HiFi DNA Assembly (New England Biolabs #2621) was used to generate the plasmids used in this study. The plasmids were then incorporated into su(Hw)attP8, attP40 or attP2 loci using the ΦC31 system.

### QUAS-mtdTomato(3xHA)

The QUAS-mtdTomato(3xHA) was amplified from UAS-myrGFP, QUAS-mtdTomato(3xHA) from the original *trans*-Tango study (*Talay et al., 2017*) using the following primers: cacggcgggcatgtcgacactagt gGTTTAAACCCAAGCTTGGATCCGGGTAATCGC and aactaggctagcggccggccttaattaaACTAGTGG ATCTAAACGAGTTTTTAAGC. First, the plasmid pUASTattB (*Bischof et al., 2007*) was digested with SpeI and the whole mix was ligated in order to reverse the orientation of the attB site. The resultant plasmid was digested with BamHI and NheI and the PCR product was cloned into the plasmid via HiFi DNA Assembly. The final plasmid was incorporated into su(Hw)attP8.

### *retro*-Tango(panneuronal)

The *retro*-Tango(panneuronal) plasmid was generated using the *trans*-Tango plasmid (*Talay et al., 2017*). The *trans*-Tango plasmid was digested with PmeI and AscI to remove the ligand and subsequently ligated to a dsDNA oligo mix containing AAACtaaGGCCGGCCcagGG and CGCGCCctgGGC CGGCCttaGTTT. The final plasmid was incorporated into attP40.

### *retro*-Tango(ligand)

The *retro*-Tango(ligand) plasmid was generated using multiple components.

The 10xUAS to flexible linker sequence from the *trans*-Tango plasmid was amplified using ttgatttt ttttttaagttggtaccCTCGAGCCTTAATTAACTGAAGTAAAG and cccagaaaggttcACTAGTATTCCCGTT ACCATTG.

The mICAM5 sequence was amplified from fly lysates (Bloomington #33062 *Nicolaï et al., 2010*) in two pieces using cgggaatactagtGAACCTTTCTGGGCGGACC & acagccatggaccGGCCACGCGCA

CTGTGAT and agtgcgcgtggccGGTCCATGGCTGTGGGTC & agttggtggcgccGGAAGATGTCAGCTG
GATAGCGAAAACC.

The P2A sequence and the farnesylated GFP (GFPfar from addgene #73014) sequence was codon optimized and synthesized by ThermoFisher. It was, then, amplified using gctgacatcttccGGCGCCA CCAACTTCTCC and ttattttaaaaacgattcatttaattaaTCAGGAGAGCACACACTTG primers.

The p10 sequence was amplified from the *trans*-Tango plasmid using tgtgctctcctgattaattaaATG AATCGTTTTTAAAATAACAAATCAATTGTTTTATAATATTCGTACG and acatcgtcgacactagtggatccg gcgcgccGTTAACTCGAATCGCTATCCAAGC.

All five PCR products were then cloned into pUASTattB[11] digested with BamHI and NheI. The final plasmid was incorporated into attP2.

## Immunohistochemistry, imaging, and image processing

Dissection of adult brains, immunohistochemistry, and imaging were performed as described in the *trans*-Tango article (*Talay et al., 2017*) with modifications to accommodate for the clearing protocol. Flies were cold anesthetized on ice and dissected in 0.05% PBST. Samples were fixed in 4%PFA/0.5% PBST for 30 min, washed four times in 0.5% PBST, blocked in heat inactivated donkey serum (5% in 0.5% PBST) for 30 min at room temperature. Samples were then treated with the primary antibody solution at 4°C for two overnights. After four washes in 0.5% PBST at room temperature, samples were treated with secondary antibody solution at 4°C for two overnights. After four washes in 0.5% PBST, samples were cleared following a previously published protocol (*Aso et al., 2014*). Reproductive system dissections were not subjected to the clearing protocol and were directly mounted on a slide (Fisherbrand Superfrost Plus, 12-550-15) using Fluoromount-G mounting medium (SouthernBiotech, 0100–01). Images were taken using confocal microscopy (Zeiss, LSM800) and were processed using the ZEN software from Zeiss. For nuclei counting, Imaris (version 9.1.2 Bitplane) was used. For cell body counting, FIJI (ImageJ2 version 2.3.0) was used and the cell bodies were counted manually. Mean number of cells ± standard deviation was reported in each figure. At least four brains for each figure were observed, a single one is represented in figures. In all images, maximum projections are shown unless otherwise stated.

The pixel intensity analysis was performed on FIJI (ImageJ2 version 2.3.0) as follows. The whole brain (for GF experiments), the central complex (for EPG experiments), or the LAL and the EB (for *retro*-Tango vs *trans*-Tango comparisons were selected via hand drawing and their integrated density was measured using the measure function). The mean pixel intensity of the background was calculated using the measure function on an unlabeled part of the brain. The pixel intensity was calculated using the following formula: pixel intensity (AU)=Integrated density of the region of interest – (Area of the region of interest X The mean pixel intensity of the background). Pixel intensities were compared using one-way ANOVA (for >2 conditions) or Student's t-test (for 2 conditions).

## Comparisons to the *Drosophila* connectome

Data from the full adult fly brain (FAFB) electron microscopy (EM) volume (*Zheng et al., 2018*) was analyzed via the hemibrain connectome (*Scheffer et al., 2020*) using the natverse suite for neuroanatomical analyses in R (*Bates et al., 2020a*). The neuprintr package (*Bates et al., 2022*) was used to query the relevant cell types that we used as the starting populations for our *retro*-Tango experiments, as well as the identity of their presynaptic partners. Synaptic strength was determined as the total number of identified synaptic connections between the starting neuron and its presynaptic partner. Neurons in which the cell bodies were not traced as part of the hemibrain connectome were excluded from our counting experiments. To plot presynaptic cells, we used neuprintr to retrieve skeletonizations of their respective EM segmentations. Since the hemibrain connectome contains only segmentations of neurons from one side of the brain, we used natverse tools for bridging registrations to mirror the presynaptic neurons across the sagittal plane to the opposite hemisphere. Briefly, skeletonizations were translated from the FAFB space to the JFRC2 template (*Jenett et al., 2012*), which contains information for translating coordinates across sagittal hemispheres. Mirrored skeletonizations were then translated back to the FAFB space and plotted alongside the unmirrored data. The R code used for analysis is available at: https://github.com/anthonycrown/retrotango, (copy archived at *Crown, 2022*).

## Acknowledgements

We acknowledge Dr. Cagney Coomer, Dr. Marnie Halpern, Dr. Jennifer Li, Dr. Karla Kaun, Daria Naumova, Dr. Drew Robson, and Dr. Rahul Trivedi for helpful discussions. We thank Dr. Alexander Fleischmann and the members of the Barnea Laboratory for critical reading of the manuscript. We are grateful to Dr. Stephen Goodwin and Susan Morton for sharing reagents. This work was supported by NIH Brain Initiative grant NIH RF1MH123213 (GB), Brown University Carney Institute for Brain Science, Suna Kıraç Fund for Brain Science (DS), Brown University Carney Institute for Brain Science, Graduate Award in Brain Science (DS) and NIH/NIDCD award F31DC019540 (AMC). Stocks obtained from the Bloomington *Drosophila* Stock Center (NIH P40OD018537) were used in this study.

## Additional information

### Funding

| Funder | Grant reference number | Author |
| --- | --- | --- |
| National Institute of Mental Health | RF1MH123213 | Gilad Barnea |
| National Institute on Deafness and Other Communication Disorders | F31DC019540 | Anthony M Crown |
| Brown University (Brown) Carney Institute for Brain Science | Suna Kirac Fund for Brain Science | Doruk Savaş Ezgi Memiş |
| Brown University (Brown) Carney Institute for Brain Science | Graduate award in brain science | Doruk Savaş |

The funders had no role in study design, data collection and interpretation, or the decision to submit the work for publication.

### Author contributions

Altar Sorkaç, Conceptualization, Resources, Data curation, Formal analysis, Supervision, Validation, Investigation, Visualization, Methodology, Writing - original draft, Project administration, Writing – review and editing; Rareş A Moşneanu, Conceptualization, Investigation, Writing - original draft, Writing – review and editing; Anthony M Crown, Conceptualization, Data curation, Software, Investigation, Writing – review and editing; Doruk Savaş, Angel M Okoro, Conceptualization, Investigation, Writing – review and editing; Ezgi Memiş, Investigation, Writing – review and editing; Mustafa Talay, Methodology, Writing – review and editing; Gilad Barnea, Conceptualization, Resources, Supervision, Funding acquisition, Writing - original draft, Project administration, Writing – review and editing

### Author ORCIDs
Altar Sorkaç ![ORCID] http://orcid.org/0000-0002-0739-6314
Gilad Barnea ![ORCID] http://orcid.org/0000-0001-6842-3454

### Decision letter and Author response
Decision letter https://doi.org/10.7554/eLife.85041.sa1
Author response https://doi.org/10.7554/eLife.85041.sa2

## Additional files

### Supplementary files
• MDAR checklist

### Data availability
The R code used for analysis is available at: https://github.com/anthonycrown/retrotango (copy archived at *Crown, 2022*).

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
