## [Editor Report]

Sorkac et al. presents a novel genetically encoded retrograde synaptic tracing method that has the potential for unbiased identification of presynaptically connected neurons. *retro*-Tango is based on the previously developed anterograde method *trans*-Tango, promising high applicability and rendering the significance of this contribution important and for some applications fundamental. The strength of the evidence is compelling and the discussion of the technique's applicability and limitations is exceptional.

---

## [Decision Letter]

**Decision letter after peer review:**

Thank you for submitting your article "*retro*-Tango enables versatile retrograde circuit tracing in *Drosophila*" for consideration by *eLife*. Your article has been reviewed by 3 peer reviewers, including P Robin Hiesinger as the Reviewing Editor and Reviewer #1, and the evaluation has been overseen by Claude Desplan as the Senior Editor. The following individual involved in the review of your submission has agreed to reveal their identity: Liqun Luo (Reviewer #2).

Essential revisions:

All three reviewers have positively evaluated the manuscript but they feel that some additional data and quantification will improve the manuscript further. The key points, as described in more detail in the reviews below, are as follows:

1. Provide more experimental validation of the specificity of ICAM5 localization to dendrites and thus retrograde specificity of the labeling technique (plausible experiments e.g. in the olfactory system).

2. Provide experimental comparisons of antero- and *retro*-Tango for the same neuron type (this will also directly address concern 1).

3. Provide quantifications throughout.

Regarding points 1 and 2: it would be useful to disclose some of the instances where the authors tried retro-T and it did not work. for example, given the lab's interest in the olfactory system, the "gold standard" would be to express the ligand in the mushroom body to test if the system can label antennal lobe PNs. Information on specific limitations might save future users time and effort.

*Reviewer #1 (Recommendations for the authors):*

The following suggestions are devised to help improve the understanding and implementation of the method:

1. Threshold analysis. The authors analyzed if the *retro*-Tango method has a specific threshold for labelling the pre-synaptic neurons and using the Giant Fiber circuit in the visual system; they compared the known pre-synaptic inputs in existing EM data to the neurons that can be identified and visualized in *retro*-Tango method. This comparison leads authors to believe that there need to be at least 15 synapses between any 2 neurons for them to be detected as synaptic partners using the *retro*-Tango method. It would be helpful to know if this threshold is specific to the visual system and/or giant fiber neurons.

2. Furthermore, the authors find that the threshold is sex-specific due to the presence of the reporter construct on the x-chromosome and therefore more efficiently expressed in males as compared to females. However, they did not try to resolve this discrepancy by generating females with 2 copies of the reporter construct. There is a possibility that there are other factors causing the differences apart from the expression strength of the reporter construct.

Additionally, if the thresholding is different for different neurons, this is critical to interpret the results. Is this the same effect as in *trans*-Tango? Some insights on this would be welcome.

3. In addition to the threshold barrier, the strength of reporter expression also depends on the time after which the flies are dissected. It would be helpful to discuss the cause of the observed differential saturation of reported signals in a different fashion in males vs females in sexually non-dimorphic circuits.

*Reviewer #2 (Recommendations for the authors):*

– A key feature of *retro*-Tango is the proposed retrograde direction. This relies entirely on the use of a fragment of a mouse cell adhesion molecule ICAM5. Although a previous study suggests selective targeting of ICAM5 in dendrites and somata but not axons when expressed in *Drosophila* neurons, given that some neuronal compartments in *Drosophila* have both pre- and postsynaptic features, and the mechanisms of neuronal polarity establishment are poorly understood, it is unclear whether the *retro*-Tango ligand with ICAM5 transmembrane domain is successfully localized and restricted to dendritic/postsynaptic sites. This is important for the interpretation of the labeling results. The authors should provide supporting evidence for the localization of *retro*-Tango ligand, for example by staining the myc tag on it, in neurons where axonal vs. dendritic compartments are well characterized. Better yet, the authors can directly test whether there is significant anterograde tracing in circuits with exclusively or predominantly unidirectional connections. For example, they could use olfactory projection neurons as starter cells; they should label olfactory receptor neurons but not mushroom body Kenyon cells.

– Most data are presented with a representative image with little quantitative information (how many samples did the authors examine, how much variation did the authors observe, etc). In their revision, the authors should provide as much quantification as they can for all the data they present.

– The best quantitative data the authors provide is the comparison with serial EM reconstruction data, leading them to conclude a threshold of 17 synapses for detecting synaptic connections by *retro*-Tango. The wording of the text gives the readers the impression of a black-and-white picture-one can detect transsynaptic labeling with 17 or more synapses, but not with fewer than 17 synapses. The reality is likely more nuanced: the labeling efficiency depends on synapse strength in addition to the number, transgene expression levels, and stochasticity in many of the steps. While "17 synapses" is a useful order-of-magnitude estimate, it is unlikely to be an absolute threshold that applies to all neurons under different experimental conditions. The authors should modify their statements taking into account the above factors.

*Reviewer #3 (Recommendations for the authors):*

I only have one request, which would not take a lot of effort: it would have been extremely valuable to be able to compare, side by side, the pattern of connectivity of retro-T with antero-T. If the authors use the same gal4 driver, and the same reporter, but they express the retro-T or antero-T ligand – are there clear, obvious differences in the labeling that they see? For example, they show the pattern of labeling with retro-T using a driver for the GF or the EPG. What does the connectivity look like if they use the EPG driver using the antero-Tango system?

I believe that this simple experiment would enormously increase the impact of this manuscript, and it should not take longer than 1 month to complete, because the authors have all the reagents from ther antero-T ligands at hand.

I would strongly recommend publication after one can compare the specificity of labeling of retro- and antero-Tango.

Further comment:

in figure 6 they express the retro-T ligand in ORNs and they see some antennal lobe neurons (projection neurons and interneurons) labeled, and they claim that this is retrograde labeling due to "reciprocal" synapses. This is not very convincing, because they got essentially the same result when they express antero-T ligand in ORNs. In sum, if they get the same type of "partner" neurons regardless of whether they express antero-T or retro-T ligands in the same neurons, this is something to worry about.

[Editors' note: further revisions were suggested prior to acceptance, as described below.]

Thank you for resubmitting your work entitled "*retro*-Tango enables versatile retrograde circuit tracing in *Drosophila*" for further consideration by *eLife*. Your revised article has been evaluated by Claude Desplan (Senior Editor) and a Reviewing Editor.

The manuscript has been improved but one specific issue should be addressed in the text based on the comment by Reviewer 3. Please add a brief discussion of the issue and a cautionary note.

*Reviewer #3 (Recommendations for the authors):*

– The new revised version is improved, and the authors have performed some of the additional experiments requested.

– I am not convinced about the data regarding the experiment where they express a forward or retrograde tango ligand on the olfactory sensory neurons. In both cases they see labeling of projection neurons in the antennal lobe. The authors claim that this is because the synapses between olfactory sensory neurons and projection neurons are reciprocal. The other scenario is that the retrograde tango ligand is not totally specific and it also labels cells in an anterograde manner.

– Overall, investigators using the retrograde version of tango will need to be cautious with the data they observe. The forward tango seems very specific to label circuits in the anterograde direction. The data from this paper indicates that the retrograde ligand may not be sufficiently specific. Probably this is due to the fact that the protein domain used to localize the retrograde tango ligand in the postsynaptic compartment of the neurons is enriched in these zones, but not exclusively localized there.

---

## [Author Response]

Essential revisions:All three reviewers have positively evaluated the manuscript but they feel that some additional data and quantification will improve the manuscript further. The key points, as described in more detail in the reviews below, are as follows:1. Provide more experimental validation of the specificity of ICAM5 localization to dendrites and thus retrograde specificity of the labeling technique (plausible experiments e.g. in the olfactory system).

To show ICAM5 localization we performed an analysis in Kenyon cells. Kenyon cell axons are in the mushroom body lobes whereas their dendrites localize to the mushroom body calyx. This analysis showed that the *retro*-Tango ligand does not localize to the mushroom body lobes as revealed by the use of the GFP-tagged Synaptotagmin1 (Figure 1—figure supplement 1).

2. Provide experimental comparisons of antero- and *retro*-Tango for the same neuron type (this will also directly address concern 1).

We conducted three experiments to compare the results of *retro*-Tango and *trans*-Tango using the same drivers. In all three experiments, *retro*-Tango and *trans*-Tango resulted in distinct signal patterns and strengths (Figure 2—figure supplement 1, Figure 6—figure supplement 1). We added the discussion of the results of these experiments in lines 175-179, 214-224, 353-360. For the EPG circuit, we also quantified the pixel intensities of the signals of these two methods in relevant regions and added the results in (Figure 2—figure supplement 1c and 1d).

3. Provide quantifications throughout.

We added quantifications to each figure, either for the number of cells labeled by *retro*-Tango or *trans*-Tango (figure legend) or for the pixel intensities (supplementary figures).

Regarding points 1 and 2: it would be useful to disclose some of the instances where the authors tried retro-T and it did not work. for example, given the lab's interest in the olfactory system, the "gold standard" would be to express the ligand in the mushroom body to test if the system can label antennal lobe PNs. Information on specific limitations might save future users time and effort.

We performed the experiment where we initiated *retro*-Tango from the Kenyon cells of the mushroom body. However, a huge part of the central brain was labeled as presynaptic to Kenyon cells, which precluded further analysis. In addition, when we initiated *retro*-Tango from OPNs, our results were inconclusive: although we did not observe labeling in the Kenyon cells, neither did we in the ORNs as would be expected. We discuss the results of these experiments where *retro*-Tango did not yield useful results in the Discussion section in lines 443-452.

Reviewer #1 (Recommendations for the authors):The following suggestions are devised to help improve the understanding and implementation of the method:1. Threshold analysis. The authors analyzed if the *retro*-Tango method has a specific threshold for labelling the pre-synaptic neurons and using the Giant Fiber circuit in the visual system; they compared the known pre-synaptic inputs in existing EM data to the neurons that can be identified and visualized in *retro*-Tango method. This comparison leads authors to believe that there need to be at least 15 synapses between any 2 neurons for them to be detected as synaptic partners using the *retro*-Tango method. It would be helpful to know if this threshold is specific to the visual system and/or giant fiber neurons.

Our calculation of a threshold of 17 synapses for observing the *retro*-Tango signal is specific to the giant fiber circuit, using 15do females heterozygous for the nuclear reporter with the particular split Gal4 driver that we used. We thank the reviewer for pointing out the need to clarify this point and to this end, we added text to the Discussion section (Lines 428-440)

2. Furthermore, the authors find that the threshold is sex-specific due to the presence of the reporter construct on the x-chromosome and therefore more efficiently expressed in males as compared to females. However, they did not try to resolve this discrepancy by generating females with 2 copies of the reporter construct. There is a possibility that there are other factors causing the differences apart from the expression strength of the reporter construct.Additionally, if the thresholding is different for different neurons, this is critical to interpret the results. Is this the same effect as in *trans*-Tango? Some insights on this would be welcome.

We performed the experiments proposed by Dr. Hiesinger, and indeed, we observed that in females homozygous for the reporter, *retro*-Tango reveals the LPLC2 neurons as presynaptic to the giant fiber. Therefore, the threshold is lower in homozygous females than in heterozygotes and is presumably closer to that in males. We have added a figure demonstrating this point (Figure 4—figure supplement 2) and discuss it in the text (Lines 289-296).

Since we did not perform a similar analysis of the threshold for *trans*-Tango, we did not add text to speculate about this in this manuscript. However, we do believe that such a threshold would also apply to *trans*-Tango and that it would depend on many factors such as the circuit of interest, the driver used, the zygosity of the reporter, the age of the animals, and the rearing conditions just like it does in *retro*-Tango.

3. In addition to the threshold barrier, the strength of reporter expression also depends on the time after which the flies are dissected. It would be helpful to discuss the cause of the observed differential saturation of reported signals in a different fashion in males vs females in sexually non-dimorphic circuits.

We would like to thank Dr. Hiesinger for this suggestion. Indeed, we performed a time-course analysis for females that are heterozygous for the reporter in the giant fiber circuit. Upon quantitative analysis of the results of this experiment, we concluded that the signal does not accumulate over time in females heterozygous for the reporter in the time frame we tested. It is conceivable that the signal accumulation is simply much slower in females heterozygous for the reporter, but the utility of the method would be reduced past this time frame. For this analysis we added Figure 3—figure supplement 3 and lines 239-242.

Reviewer #2 (Recommendations for the authors):– A key feature of *retro*-Tango is the proposed retrograde direction. This relies entirely on the use of a fragment of a mouse cell adhesion molecule ICAM5. Although a previous study suggests selective targeting of ICAM5 in dendrites and somata but not axons when expressed in *Drosophila* neurons, given that some neuronal compartments in Drosophila have both pre- and postsynaptic features, and the mechanisms of neuronal polarity establishment are poorly understood, it is unclear whether the *retro*-Tango ligand with ICAM5 transmembrane domain is successfully localized and restricted to dendritic/postsynaptic sites. This is important for the interpretation of the labeling results. The authors should provide supporting evidence for the localization of *retro*-Tango ligand, for example by staining the myc tag on it, in neurons where axonal vs. dendritic compartments are well characterized. Better yet, the authors can directly test whether there is significant anterograde tracing in circuits with exclusively or predominantly unidirectional connections. For example, they could use olfactory projection neurons as starter cells; they should label olfactory receptor neurons but not mushroom body Kenyon cells.

As Dr. Luo suggested, we performed an analysis of the localization of the *retro*-Tango ligand in Kenyon cells where the axonal and dendritic compartments are distinct. Indeed, this analysis demonstrated that the *retro*-Tango ligand does not localize to the axons of the Kenyon cells as revealed by the use of the GFP-tagged Synaptotagmin1 (Figure 1—figure supplement 1).

Further, we performed three experiments in which we compared the signals of *retro*-Tango and *trans*-Tango using the same starter neurons. In all three cases, we observed distinct signal patterns and strengths with the two systems (Figure 2—figure supplement 1, Figure 6—figure supplement 1). We discussed the results of these experiments in lines 175-179, 214-224, 353-360. For the EPG circuit, we also quantified the pixel intensities of the signals of these two methods in relevant regions and added the results in (Figure 2—figure supplement 1c and 1d).

As to the experiment using the olfactory projection neurons suggested by Dr. Luo, our results are inconclusive. While, as expected, we did not observe signal in the Kenyon cells when the *retro*-Tango was initiated from the GH146-expressing OPNs, neither did we observe the expected signal in ORNs. We discussed this in the Discussion section, lines 442-451.

– Most data are presented with a representative image with little quantitative information (how many samples did the authors examine, how much variation did the authors observe, etc). In their revision, the authors should provide as much quantification as they can for all the data they present.

We thank Dr. Luo for this important comment. We corrected this throughout the manuscript either by pixel intensity analysis for direct comparisons or by counting the number of cells revealed by *retro*-Tango or *trans*-Tango. We believe that consequently the manuscript has substantially improved.

– The best quantitative data the authors provide is the comparison with serial EM reconstruction data, leading them to conclude a threshold of 17 synapses for detecting synaptic connections by *retro*-Tango. The wording of the text gives the readers the impression of a black-and-white picture-one can detect transsynaptic labeling with 17 or more synapses, but not with fewer than 17 synapses. The reality is likely more nuanced: the labeling efficiency depends on synapse strength in addition to the number, transgene expression levels, and stochasticity in many of the steps. While "17 synapses" is a useful order-of-magnitude estimate, it is unlikely to be an absolute threshold that applies to all neurons under different experimental conditions. The authors should modify their statements taking into account the above factors.

We thank Dr. Luo for this important comment. We substantially revised our description of this analysis and our discussion of the results. We are grateful to Dr. Luo for bringing this to our attention because indeed painting a black-and-white picture was not our intention but in retrospect our original description could have led the readers to this conclusion. We believe that our revised text paints a much more nuanced picture that is more consistent with our intention. To this end, we added lines 293-296 and 428-440.

Reviewer #3 (Recommendations for the authors):I only have one request, which would not take a lot of effort: it would have been extremely valuable to be able to compare, side by side, the pattern of connectivity of retro-T with antero-T. If the authors use the same gal4 driver, and the same reporter, but they express the retro-T or antero-T ligand – are there clear, obvious differences in the labeling that they see? For example, they show the pattern of labeling with retro-T using a driver for the GF or the EPG. What does the connectivity look like if they use the EPG driver using the antero-Tango system?I believe that this simple experiment would enormously increase the impact of this manuscript, and it should not take longer than 1 month to complete, because the authors have all the reagents from ther antero-T ligands at hand.I would strongly recommend publication after one can compare the specificity of labeling of retro- and antero-Tango.

We thank the reviewer for this suggestion, and we think that these experiments enhanced our manuscript significantly. We performed the *trans*-Tango experiments as the reviewer suggested for the EPG and GF circuits. Initiating trans-Tango from the GF resulted in a completely different pattern than *retro*-Tango did. We added Figure 2—figure supplement 1a for *trans*-Tango results and discussed them in lines 175-179. When we initiated *trans*-Tango from the EPG circuit, we observed a similar but distinct pattern compared to that of *retro*-Tango. To quantify the differences, we performed pixel intensity analysis in the ellipsoid body and the lateral accessory lobes. For the EPG circuit we added Figure 2—figure supplement 1b,c and d, and we discussed the results of these experiments in lines 214-224.

Further comment:in figure 6 they express the retro-T ligand in ORNs and they see some antennal lobe neurons (projection neurons and interneurons) labeled, and they claim that this is retrograde labeling due to "reciprocal" synapses. This is not very convincing, because they got essentially the same result when they express antero-T ligand in ORNs. In sum, if they get the same type of "partner" neurons regardless of whether they express antero-T or retro-T ligands in the same neurons, this is something to worry about.

To address the reviewers concerns about using *retro*-Tango in circuits with reciprocal synapses, we performed *trans*-Tango experiments using the same Or67d-Gal4 driver. Although we observed a similar pattern to that of retro-Tango, we also observed obvious differences. We especially noted that the mediolateral antennal lobe tract, clearly visible in *trans*-Tango experiments, was not marked as presynaptic using *retro*-Tango, showcasing that the two systems lead to distinct results. For this experiment we added Figure 6—figure supplement 1 and discussed the results in lines 353-361.

[Editors' note: further revisions were suggested prior to acceptance, as described below.]

The manuscript has been improved but one specific issue should be addressed in the text based on the comment by Reviewer 3. Please add a brief discussion of the issue and a cautionary note.Reviewer #3 (Recommendations for the authors):– The new revised version is improved, and the authors have performed some of the additional experiments requested.– I am not convinced about the data regarding the experiment where they express a forward or retrograde tango ligand on the olfactory sensory neurons. In both cases they see labeling of projection neurons in the antennal lobe. The authors claim that this is because the synapses between olfactory sensory neurons and projection neurons are reciprocal. The other scenario is that the retrograde tango ligand is not totally specific and it also labels cells in an anterograde manner.– Overall, investigators using the retrograde version of tango will need to be cautious with the data they observe. The forward tango seems very specific to label circuits in the anterograde direction. The data from this paper indicates that the retrograde ligand may not be sufficiently specific. Probably this is due to the fact that the protein domain used to localize the retrograde tango ligand in the postsynaptic compartment of the neurons is enriched in these zones, but not exclusively localized there.

To address the Reviewer’s comments, we added the following text to the discussion:

“Finally, in the Or67d circuit, we attribute the similarity between the *retro*-Tango and *trans*-Tango signals to the known reciprocal connections between ORNs, OPNs and LNs (Horne et al., 2018). The clear distinction between the signals in the other two circuits (EPG and GF) supports this interpretation. That said, it is not inconceivable that with certain drivers in certain circuits some false positive signal might be observed in the anterograde direction if the ligand localizes outside the postsynaptic membrane. However, even if the *retro*-Tango ligand is only enriched in the postsynaptic membrane and not exclusively targeted there, one would expect the levels of the ligand at the presynaptic sites to be minimal and mostly below the threshold to activate the Tango cascade. Nonetheless, users should be cognizant of the possibility of anterograde labeling.”